# The Central Domain of MCPH1 Controls Development of the Cerebral Cortex and Gonads in Mice

**DOI:** 10.3390/cells11172715

**Published:** 2022-08-31

**Authors:** Yaru Wang, Wen Zong, Wenli Sun, Chengyan Chen, Zhao-Qi Wang, Tangliang Li

**Affiliations:** 1State Key Laboratory of Microbial Technology, Shandong University, Qingdao 250100, China; 2Leibniz Institute on Aging—Fritz Lipmann Institute (FLI), 07745 Jena, Germany; 3Faculty of Biological Sciences, Friedrich-Schiller University of Jena, 07743 Jena, Germany; 4Department of Pathology and Pathophysiology, School of Basic Medical Sciences, Hangzhou Normal University, Hangzhou 311121, China

**Keywords:** microcephaly, MCPH1, central domain, brain development, gonad development

## Abstract

*MCPH1* is the first gene identified to be responsible for the human autosomal recessive disorder primary microcephaly (MCPH). Mutations in the N-terminal and central domains of MCPH1 are strongly associated with microcephaly in human patients. A recent study showed that the central domain of MCPH1, which is mainly encoded by exon 8, interacts with E3 ligase βTrCP2 and regulates the G2/M transition of the cell cycle. In order to investigate the biological functions of MCPH1’s central domain, we constructed a mouse model that lacked the central domain of MCPH1 by deleting its exon 8 (designated as *Mcph1*-Δe8). *Mcph1*-Δe8 mice exhibited a reduced brain size and thinner cortex, likely caused by a compromised self-renewal capacity and premature differentiation of *Mcph1*-Δe8 neuroprogenitors during corticogenesis. Furthermore, *Mcph1*-Δe8 mice were sterile because of a loss of germ cells in the testis and ovary. The embryonic fibroblasts of *Mcph1*-Δe8 mice exhibited premature chromosome condensation (PCC). All of these findings indicate that *Mcph1*-Δe8 mice are reminiscent of MCPH1 complete knockout mice and *Mcph1*-ΔBR1 mice. Our study demonstrates that the central domain of MCPH1 represses microcephaly, and is essential for gonad development in mammals.

## 1. Introduction

Autosomal recessive primary microcephaly (MCPH) is a rare heterogenous neurodevelopmental disorder that is characterized by a pronounced reduction in the brain size. It is caused by defects in neuroprogenitors during neurodevelopment, and has an incidence of 1:30,000 to 1:250,000 in live births in the general population [1,2,3]. MCPH patients exhibit a small brain with simplified gyri, and by marked reduction in the cerebral cortex, although the brain architecture is grossly normal [4]. Currently, mutations in 29 genes have been identified as causes of MCPH [5,6,7]. *MCPH1* (*or BRIT1*) was the first gene reported to be causative for primary microcephaly type 1 (MCPH1, OMIM251200) [1]. In addition to brain developmental abnormalities, MCPH mutations cause premature chromosome condensation (PCC) syndrome (OMIM 606858) [8,9]. MCPH patient cells show defective chromosome condensation, with a high proportion of prophase-like cells (PLCs) in late G2-phase with delayed de-condensation post-mitosis [8,10].

The *MCPH1* gene encodes a multifunctional protein that plays an important role in chromosome condensation, DNA damage response (DDR), cell cycle progression, chromatin remodeling and tumorigenesis [11,12]. These functions enable MCPH1 to play an important role in brain development, gonad formation and tumorigenesis [7,13,14]. MCPH1 contains three functional domains: the N-terminal BRCT domain (BRCT1) is necessary for prevention of PCC and contributes to the centrosome localization of MCPH1 [15,16]; the two C-terminal BRCT domains (BRCT2 and BRCT3) bind to phosphorylated proteins in DDR, and are crucial for its localization to DNA damage sites by interacting with γH2AX after ionizing radiation [17,18]. The central domain of MCPH1 (amino acids 381–435) is primarily responsible for the condensin II–MCPH1 interaction [19].

Currently, numerous loss-of-function mutations and variants of *MCPH1* have been associated with MCPH [12,20]. Most missense mutations of *MCPH1* are located in exons 2 and 3 that encode the N-terminal BRCT domain. In accordance to human MCPH1, complete knockout of *Mcph1* (*Mcph1*-del) by deletion of exons 4 and 5 in mice results in primary microcephaly, recapitulating human MCPH1 [21]. Mechanistically, MCPH1 loss reduces Chk1 in the centrosome, leading to aberrant Cdk1 activation and premature mitotic entry of neuroprogenitors [21]. Via this mechanism, MCPH1 regulates the division mode of neuroprogenitors. Of note, mice expressing MCPH1 that is missing the first BRCT domain at the N terminus (*Mcph1*-ΔBRCT1, or *Mcph1*-ΔBR1) reproduce the primary microcephaly phenotype as seen in MCPH1 knockout mice and MCPH patients [13].

Recently, novel MCPH1 mutations have been found to be linked to disruptions in the MCPH1 middle domain. For example, two heterozygous missense mutations (C.982G > A and C.1273T > A) in exon 8 of MCPH1 were found in microcephaly individuals of a Saudi family [22]. A recent finding reports a novel frameshift deletion mutation c.1254delT in exon 8 of the *MCPH1* gene which disrupts the conserved central domain of MCPH1 [23]. These findings suggest an important function of the central domain of MCPH1 in brain development. Our recent molecular biological study conducted in human 293T and Hela cells showed that MCPH1, via its central domain, modulated the dimerization of βTrCP2 to regulate Cdc25A turnover [24,25]. Thus, MCPH1 controls the G2/M transition and determines the mitotic fate of neuroprogenitors [24]. In the current study, we aimed to investigate the physiological function of the central domain of MCPH1, and generated a knockout mouse model lacking exon 8 of *Mcph1* (designated as *Mcph1*-Δe8), which expresses a truncated MCPH1 without its central domain. Our analysis showed that the central domain of MCPH1 is essential for corticogenesis and gonad development in mice.

## 2. Materials and Methods

### 2.1. Mice and Mating Scheme

MCPH1’s central domain conditional knockout mouse (*Mcph1*^flox-e8/flox-e8^) was produced by a commercial service provided by Cyagen Biosciences Inc., Suzhou, China. The targeting vector was designed as follows: homology arms (a 2.9-kb 5′ arm and 2.6-kb 3′ arm) were generated using BAC clones from the C57BL/6J RPCI-23 BAC library; the 5′ LoxP site was inserted into intron 7 of *MCPH1* together with a self-excision Neo cassette flanked by Rox sites that can be cleaved by the testes-specific Dre recombinase (SDA–Neo–SDA–LoxP cassette) (Figure 1A); the second LoxP site (3′ LoxP) was inserted into intron 8. The negative selection marker diphtheria toxin A (DTA) cassette was placed upstream of the 5′ homology arm. The linearized targeting vector was introduced into Cyagen’s proprietary TurboKnockout ES cells (on a C57BL/6N background) by electroporation. Homologous recombinant clones were isolated using positive (neomycin) and negative (DTA) selection. There were 188 neo-resistant ES clones that were analyzed by Southern blotting in order to identify the gene targeting events. Three probes for Southern blotting were designed to screen the correctly targeted ES clones. The probe 1 (P1) in intron 7 within the 5′ homology arm detected a fragment of 8.44 kb in wild-type (WT) allele and 3.39 kb in targeted (Tg) allele after KpnI digestion (Figure 1A). The probe 2 (P2) was located in intron 8, which detected a fragment of 13.91 kb in WT allele and 10.76 kb in Tg allele after AvrII digestion (Figure 1A). The probe Neo detected a fragment of 9.06 kb after genomic DNA digestion with KpnI, which was used to rule out any random insertion of the targeting vector into other parts of the genome (Appendix A). In total, six ES clones were confirmed to have correct gene targeting events. The targeted ES cell clone 1B1 was introduced into blastocysts by microinjection and then surgically transferred into pseudo-pregnant (surrogate) mothers in order to generate chimeric offspring, which were further crossed with C57BL/6N females to generate heterozygous floxed mice (*Mcph1*^flox-e8/+^). The *Mcph1*^flox-e8/flox-e8^ mouse was bred with the EIIa-Cre transgenic mouse to remove the exon 8 and to generate a conventional central domain knockout mouse for MCPH1 (*Mcph1*-Δe8) (see the Results section; Figure 1 and Appendix A).

The *MCPH1* alleles were genotyped using the following primers: F1: GGTCTGAGTAATGACCACAGGTTC; R1: GTGGGTAAACACAACTCATCCTTC; R2: TGGGTACTCCTGCTAGCCTC (wild-type allele, 193 bps; flox allele, 327 bps; and exon 8 deletion allele, 543 bps). EIIa-Cre transgene was genotyped by the primers Cre1, 5′-CATATTGGCAGAACGAAAACGC-3′; and Cre-2, 5′-CCTGTTTCACTATCCAGGTTACGG-3′; the amplicon size was 413 bps. 

Mouse strains, including *Mcph1*^flox-e8/flox-e8^, *Mcph1*-Δe8 and EIIa-Cre mice, were maintained under specific pathogen-free conditions at the animal facility of Shandong University, Qingdao, P. R. China. Animal care and experiments were performed in accordance with the ethics committee’s guidelines (License number: SYDWLL-2022-064). 

### 2.2. Cell Culture 

Mouse embryonic fibroblasts (MEFs) were isolated from E13.5 embryos following previously published protocols [26]. Neuroprogenitors were isolated from E14.5 embryonic cortex tissues and maintained in the following neural stem cell medium: DMEM/F12 (Gibco) supplemented with B-27 (Gibco), penicillin/streptomycin (Gibco), 10 ng/mL EGF (Peprotech) and 20 ng/mL bFGF (Peprotech) [27]. All of the cell cultures were maintained at 37 °C in a CO_2_ incubator. 

### 2.3. RNA Extraction and RT-PCR Analysis

Total RNAs were extracted from cells and mouse tissues (cerebral cortex) using an RNA extraction kit (Accurate Biotechnology, Changsha, China). cDNA was synthesized using PrimeScript II 1st Strand cDNA Synthesis Kit (Takara, Kusatsu, Japan). RT-PCRs for different fragments of *Mcph1* were conducted using PrimerSTAR^®^ HS premix (Takara, Kusatsu, Japan). The primers used to characterize different fragments of *Mcph1* are listed in Appendix A. PCR products were isolated with the Gel Extraction Kit (Sangon Biotech, Shanghai, China) and sequenced through services provided by Tsingke.

### 2.4. Histological Analysis

For paraffin sections, mouse brains were fixed with 4% paraformaldehyde at 4 °C for 24–48 h. Testicular and ovarian tissues were fixed with Bouin’s solution for 24 h. Tissue samples were further processed and embedded in paraffin. Sections of 4 μm were used in this study. The sections were stained with hematoxylin and eosin solutions. The histological images were scanned and processed with an Olympus VS200 microscope.

### 2.5. Immunofluorescence (IF) Staining on Cells and Brain Sections

For IF analysis of cells, primary MEF cells were fixed with 4% PFA for 15 min at room temperature. Then, MEFs were stained with pS10-histone 3 antibody (1:400, 9706S, Cell Signaling, Danvers, MA, USA) in order to investigate the mitotic cells. At least 250 prophase cells from each cell line were counted manually.

For IF in brain tissues, embryonic brains at indicated developmental stages were fixed with 4% PFA at 4 °C for 24–48 h, then transferred to 30% sucrose. Brains were then embedded with OCT (Thermo Fisher, Waltham, MA, USA) and stored at −80 °C. A 10-micrometer section was used for cryosection. For immunostaining, a previously published protocol was followed [27]. The following primary antibodies were used: rabbit anti-Sox2 (1:400, Ab97959, Abcam, Cambridge, UK), rabbit anti-Tbr2 (1:400, Ab183991, Abcam, Cambridge, UK), rabbit anti-Ki67 (1:400, 9129S, Cell Signaling), rabbit anti-cleaved caspase-3 (Asp175) (1:400, 9579S, Cell Signaling, Danvers, MA, USA), mouse anti-pS10-H3 (1:400, 9706S, Cell Signaling, Danvers, MA, USA) and rat anti-pS28-H3 (1:400, Ab10543, Abcam, Cambridge, UK). The following secondary antibodies were used: Goat Anti-Mouse IgG Alexa Fluor^®^ 594 (1:1000, Ab150113, Abcam, Cambridge, UK), Goat Anti-Rabbit IgG Alexa Fluor^®^ 594 (1:1000, Ab150080, Abcam, Cambridge, UK), and Goat Anti-Rat IgG Alexa Fluor^®^594 (1:1000, Ab150160, Abcam, Cambridge, UK). The antibodies used for IF analysis are summarized in Appendix A).

All of the IF images from cells and tissues were captured with OLYMPUS cellSens software (Standard Version, Shinjuku City, Tokyo, Japan) on an Olympus microscope BX53 installed with a DP80 camera. The images were further analyzed and processed with OLYMPUS OlyVIA software (Version 3.2, Shinjuku City, Tokyo, Japan).

### 2.6. Western Blotting

In order to detect MCPH1 expression, cell or tissue samples were lysed with RIPA buffer supplements with protease/phosphatase inhibitors (APExBIO). An amount of 40–60 ug total protein was separated with SDS-PAGE gels (10%), and further transferred onto PVDF membranes. The primary antibodies used for this study were the following: rabbit anti-MCPH1 (1:1000, Cell Signaling, Danvers, MA, USA); and mouse anti-β-actin (1:10,000, Sigma-Aldrich, St. Louis, MO, USA). For the DNA damage assay, MEFs were treated with hydroxyurea (HU, 2 mM) for 3 h, and then recovered for 0, 3 and 6 h. Protein samples were harvested at different time-points. In order to investigate the DNA repair dynamics, the following antibodies were used: mouse anti-phospho histone H2A.X (Ser139) (1:2000, 05-636, EMD Millipore, Burlington, MA, USA), rabbit anti-phospho Chk1 (Ser317) (1:1000,12302S, Cell Signaling, Danvers, MA, USA) and mouse anti-β-actin (1:10,000, A5441, Sigma-Aldrich, St. Louis, MO, USA). The secondary antibodies used were: HRP-conjugated goat anti-rabbit IgG and HRP-conjugated goat anti-mouse IgG (1:2000; Proteintech, Tokyo, Japan). The blotting result was developed using BeyoECL Plus substrates (Beyotime, P0018S, Shanghai, China) and quantified with Evolution-Capt Solo S 17.00 software. The antibodies used for WB are summarized in Appendix A).

### 2.7. Statistical Analysis 

For the quantitative analysis conducted in this study, at least 3 biological samples from each genotype were used. Unpaired Student’s *t*-test was employed. The statistical analysis in this study was performed with GraphPad Prism (Ver 6.00, GraphPad Software, San Diego, CA, USA) and graphed with the format of mean ± SD. Statistical significance between genotypes was used as follows: n.s, *p* > 0.05; *, *p* < 0.05; **, *p* < 0.01; ***, *p* < 0.001.

## 3. Results

### 3.1. Generation of Mcph1-Δe8 Mice

The central domain of mouse MCPH1 is mainly encoded by exon 8 of the *Mcph1* gene. Thus, we generated an Mcph1 exon 8 conditional knockout mouse by introducing LoxP sites that flanked exon 8 (Figure 1A, please see the Materials and Methods section for details). In order to produce conventional knockout mice with MCPH1’s central domain deletion, *Mcph1*^flox-e8/+^ mice were further crossed with EIIa-Cre mice to delete exon 8 of *Mcph1* in the germline. Heterozygous *Mcph1*^Δe8/+^ mice were further intercrossed to generate *Mcph1*^Δe8/Δe8^ mice (designated as *Mcph1*-Δe8) (Appendix A). In order to confirm the successful generation of the *Mcph1*-Δe8 mouse, we performed PCR genotyping and found that the Δe8 mutant allele (Δe8) produced an expected 543-base-pair PCR product (Appendix A). RT-PCR experiments revealed the deletion of exon 8 in *MCPH1* transcripts in *Mcph1*-Δe8 mouse embryos (Figure 1B, primer set E8-E8as), which was further confirmed by sequencing (Figure 1C). Of note, the mouse *Mcph1* exon 8 is 1128 bps, deletion of which resulted in in-frame deletion of the central domain in MCPH1. Western blotting using an MCPH1 antibody (D38G5, #4120, Cell Signaling Technology) that recognizes the MCPH1’s central domain confirmed that *Mcph1*-Δe8 indeed was missing this middle domain (Figure 1D). All of these data indicate that the *Mcph1*-Δe8 mouse was successfully generated.

### 3.2. Mcph1-Δe8 Mice Develop Microcephaly

Intercross of heterozygous *Mcph1*-Δe8 mice (*Mcph1*^Δe8/+^) generated homozygous *Mcph1*-Δe8 mice at birth with normal Mendelian ratios (data not shown). Interestingly, we noticed that *Mcph1*-Δe8 mice had a smaller brain as well as a reduction in the brain weight as compared to control littermates at postnatal day 0 (P0) (Figure 2A,B), which is similar to *MCPH1* complete knockout mice [21] and *Mcph1*-ΔBR1 mice [13]. In addition to the cortex, the middle brain of *Mcph1*-Δe8 mice at P0 seemed smaller than that of control mice (Figure 2A). Histological analysis of coronal sections of P0 brains revealed a thinner cerebral cortex in *Mcph1*-Δe8 as compared to control mice (Figure 2C,D), indicating that *Mcph1*-Δe8 mice are microcephalic.

### 3.3. Mcph1-Δe8 Neuroprogenitors Have Self-Renewal Defect

In order to study the reasons behind the primary microcephaly phenotype of *Mcph1*-Δe8 mice, we analyzed the embryonic cortex at the middle stage of neurogenesis, namely E15.5. By immunostaining with antibodies against Sox2, a marker for radial glial cells (RGCs) in the ventricular zone (VZ), and Tbr2, a marker for intermediate progenitors (IPs) in the subventricular zone (SVZ), we found less Sox2-positive (Sox2^+^) cells in the *Mcph1*-Δe8 cortex compared to controls (Figure 3A,B); however, we found a normal number of Tbr2-positive (Tbr2^+^) cells in the SVZ (Figure 3A,B), indicating that RGCs, but not intermediate progenitors, are affected by the mutation. 

In order to test whether the fewer Sox2^+^ RGCs observed were caused by an increase in apoptosis, we performed immunofluorescence staining for cleaved-Caspase3 (a classical assay to detect apoptotic cells) and found there was no obvious difference in the apoptotic index between the control’s and *Mcph1*-Δe8 genotype’s cerebral cortex tissues (Appendix A). We next analyzed the proliferation of the neuroprogenitors. In vivo pulse labeling (1 h) by EdU incorporation into E15.5 embryos revealed that the ratio of EdU^+^ cells in the VZ and SVZ of the *Mcph1*-Δe8 cortex was significantly reduced as compared to control littermates (Figure 4A,B). We next examined whether the low number of EdU-positive cells affects cell division by measuring the number of mitotic cells after immunostaining against mitosis maker pS28-H3 (Figure 4A). Indeed, the number of pS28-H3-positive cells was much less in the *Mcph1*-Δe8 cortex compared to control littermates (Figure 4B). Thus, the middle domain of MCPH1 is required for the self-renewal and maintenance of neuroprogenitors [22,23,24], but dispensable for apoptosis.

### 3.4. Mcph1-Δe8 Neuroprogenitors Are Prone to Differentiate Prematurely

Previously, we found that a complete loss of MCPH1 drives the premature differentiation of neuroprogenitors [21]. In order to investigate whether the reduced number of neuroprogenitors was linked to their premature differentiation, we performed an in vivo differentiation assay by measuring the cell cycle exit index of RGCs in the neocortex [21]. To this end, E14.5 pregnant mice were injected with EdU, and embryonic brains were collected at E15.5 for immunostaining with Ki67 antibody. We quantified the cell cycle exit index (EdU^+^Ki67^−^ cells in the total EdU^+^ cells) of the mutant and control neocortex (Figure 4C,D) and found that the *Mcph1*-Δe8 neocortex had a much higher cell cycle index than that of the control. This finding indicates that the deletion of *Mcph1* exon 8 drives the premature differentiation of neuroprogenitors. 

### 3.5. Mcph1-Δe8 Mutation Renders Cells to PCC and Malfunctional DDR

In order to investigate whether *Mcph1*-Δe8 mice also have the PCC phenotype characteristics of cells from MCPH patients, we isolated primary MEF cells. These cells were stained with the pS10-H3 antibody that labels metaphase cells and then counter-stained with DAPI to visualize chromatin/chromosomes. The PCC cells were judged by condensed chromosomes but lacking pS10-H3 signals (Figure 5A). The *Mcph1*-Δe8 primary MEF cells had a significantly higher percentage of PCC than controls (Figure 5B, 77% in *Mcph1*-Δe8 vs. 1.75% in control). 

Human cells with siRNA knockdown of MCPH1, or MCPH patient lymphoblast cells, exhibit a defective DDR that involves the ATR-Chk1-mediated cell cycle checkpoint [28,29]. We next investigated if *Mcph1*-Δe8 primary MEF cells had altered Atr-Chk1 signaling using Western blotting. Hydroxyurea (HU) inhibits DNA synthesis, and has been widely used to investigate Atr-Chk1 -mediated DDR [26]. To this end, we treated the control and *Mcph1*-Δe8 MEFs with or without HU, and analyzed the expressions of p-Chk1 and γH2AX at different time-points. p-Chk1 levels indicate an activation of Atr signaling, while γH2AX levels mark DNA damage accumulation in the cells [26]. We found that *Mcph1*-Δe8 MEF cells had higher basal as well as HU-induced levels of p-Chk1 and γH2AX compared to those of controls (Figure 5C,D), indicating that these mutant cells have malfunctional Atr signaling.

### 3.6. Defects in Gonad Development in Mcph1-Δe8 Mice

We crossed *Mcph1*-Δe8 mice, both males and females, with wild-type mice for 4 months, but all mutant mice failed to produce any offspring (data not shown), indicating that *Mcph1*-Δe8 mice were infertile, similarly to the sterility of *Mcph1*-del and *Mcph1*-ΔBR1 mice [13,21]. Macroscopically, the *Mcph1*-Δe8 testis was significantly smaller than that of the control (Figure 6A). The ratio of testis weight (TW) to body weight (BW) was also significantly smaller in *Mcph1*-Δe8 mice (Figure 6A). Histological analysis after H&E staining showed that *Mcph1*-Δe8 testes were devoid of round spermatids and elongated spermatids, but still possessed some pachytene spermatocytes within the seminiferous tubules (Figure 6B). In addition, we performed H&E staining of the epididymis and did not detect any spermatozoa-filled tubules in the *Mcph1*-Δe8 epididymis (Figure 6B). The diameters of *Mcph1*-Δe8 seminiferous tubules in testes were smaller (Figure 6C). A significantly higher frequency of vacuolized seminiferous tubules was found in *Mcph1*-Δe8 testes (Figure 6D). These data suggest that spermatogenesis in *Mcph1*-Δe8 mice was arrested at an early stage of development. In female *Mcph1*-Δe8 mice, the size of the ovary was smaller and the uterine wall was thinner compared to controls (Figure 6E). H&E staining revealed that ovarian follicles were invisible in *Mcph1*-Δe8 female mice (Figure 6F). We conclude that the central domain of MCPH1 is essential for gonad development.

## 4. Discussion

In humans, MCPH1 mutations cause primary microcephaly. Genetic studies identify that many mutations and genetic variants of MCPH1 are prominently located in the N-terminal and central domains [7,20,22,23,30]. We previously generated MCPH1 conventional knockout mice (*Mcph1*-del) [21], MCPH1 N-terminal BRCT domain knockout mice (*Mcph1*-ΔBR1) [13], and MCPH1 fetal brain specific knockout mice [21,31], and revealed that the whole MCPH1 protein and the first BRCT domain at the MCPH1 N’ terminus are essential for neuroprogenitor self-renewal and differentiation during corticogenesis. The first BRCT domain of MCPH1 is required for its centromere localization, which has an impact on choices of symmetric and asymmetric division of neuroprogenitors. Thus, the altered cell division mode in neuroprogenitors from *Mcph1*-del and *Mcph1*-ΔBR1 embryonic brains is responsible for the microcephaly phenotype in these models [21,24]. 

Recently, we found that MCPH1, via its central domain, directly interacts with βTrCP2 to promote its activity to degrade Cdc25A during the G2/M transition in cell cycle control. Ectopic expression of βTrCP2 or the Cdc25A knockdown remedied the premature differentiation of MCPH1-deficient neuroprogenitors [24]. Therefore, a stabilization role of the MCPH1 central domain on βTrCP2 is presumably important for brain development. We showed here, using a mouse model in which the central domain of MCPH1 is deleted (by deleting exon 8), that the central domain of MCPH1 is required for preventing microcephaly, and maintains a proper balance of the self-renewal and differentiation capacity of neuroprogenitors during brain development. *Mcph1*-Δe8 mice show a reduction in cortical thickness that is likely due to impaired proliferation and premature differentiation of neuroprogenitors during the early stages of embryonic neurogenesis, ultimately leading to a reduction in the neuroprogenitor pool and a small brain size. 

In addition to brain developmental defects, *Mcph1*-Δe8 mice show severe gonad atrophy in both males and females, similarly to *MCPH1* complete knockout and *Mcph1*-ΔBR1 mice [13,14,21]; this indicates that the central domain of MCPH1 is essential for gonadal development. It might be surprising that all of these *Mcph1* mutant mice are always associated with defects in gonad development and infertility. It strongly suggests that at least in mice, MCPH1 is critical for the normal development of the reproductive system [13,14,21]. Currently, however, there is no case reports of MCPH patients with testicular or ovarian atrophy. How MCPH1 regulates spermatogenesis or ovary formation is currently unclear. One possible explanation is that defective DNA damage repair (as judged by a high level of p-Chk1 and γH2AX shown in *Mcph1*-Δe8 MEFs) may contribute to infertility phenotypes in *Mcph1*-Δe8 testes and ovaries. It was found that the recruitment of DNA repair proteins BRCA2-RAD51 was impaired in MCPH1 mutant mice, leading to meiosis arrest and apoptosis of spermatocytes, as well as a completely loss of pachytene and post-meiosis spermatocytes [14]. 

The central domain of MCPH1 has been shown to interact with condensin II, which is believed to be responsible for chromosome condensation [16,19]. Similarly to *Mcph1*-deficient [14,21] and *Mcph1*-ΔBR1 cells [13], *Mcph1*-Δe8 MEF cells exhibit a high incidence of PCC, indicating that the central domain is involved in the regulation of pre-mitotic chromosomal states. Of note, previous studies found that *Mcph1*-ΔBR1 MEFs have around 40% of PCC, while complete knockout (*Mcph1*-del) MEFs have around 30% PCC [13,21]. The difference in PCC index in each mouse line could be caused by an MEF isolation process, cell culture conditions, or molecular markers used to define metaphase cells.

In summary, we demonstrated that analogously to *Mcph1*-complete knockout [14,21] and *Mcph1*-ΔBR1 [13] mice, the deletion of the central domain of MCPH1 in mice recapitulates phenotypes of MCPH patients, i.e., primary microcephaly and the PCC. The three MCPH1 mutant mouse lines generated in our lab [13,21], together with three additional MCPH1 mutants produced by other research groups [14,32,33], allowed us to conclude that the decisive function of MCPH1 in brain development and fertility is located in the N-terminal and the central domains of MCPH1. Since MCPH1 has no enzymatic activity, it is plausible that MCPH1 participates in various cellular activities through interactions with different partners. For example, MCPH1 interacts with βTrCP2 through its central domain and participates in the regulation of the G2/M cell cycle checkpoint [24]. Therefore, finding proteins that interact with specific domains of MCPH1 could provide further hints to decipher the biological functions of MCPH1 in all of these physiological and developmental processes.

## Figures and Tables

**Figure 1 cells-11-02715-f001:**
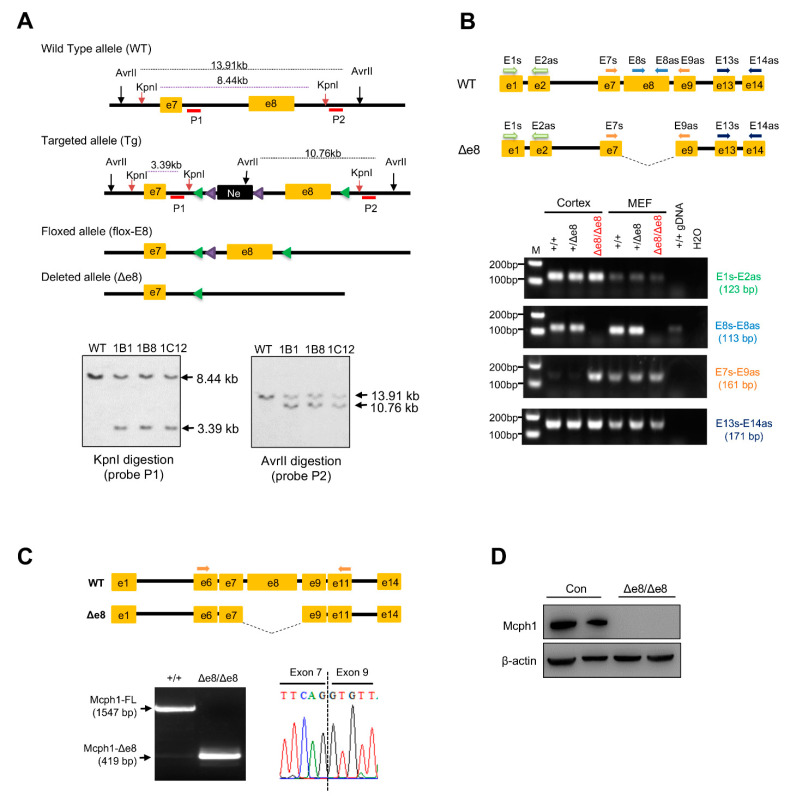
Generation of the MCPH1 central domain knockout mouse (*Mcph1*-Δe8). (**A**) Gene targeting strategy to generate MCPH1 central domain knockout mice. Exon 8 of *Mcph1* that encodes the MCPH1 central domain was floxed by two LoxP sites. The expected alleles before and after gene targeting are shown. The expected sizes of wild-type allele (WT), targeted allele (Tg), floxed allele (Flox) and exon 8 deleted allele, after respective enzyme digestion in Southern blotting, are shown. The locations of the Southern blotting probes (P1 for the 5′ homology arm; P2 for the 3′ homology arm) are marked under their respective alleles. Lower panel: Southern blotting to identify the correct homologous recombination events in ES cell clones after digestion with the indicated restriction enzymes and hybridization with the indicated probes. (**B**) RT-PCR analysis to validate exon 8 deletion in the *Mcph1* transcript of the *Mcph1*-Δe8 cortex and MEFs. Top panel: schematic presentation of localization of PCR primers in the WT and *Mcph1*-Δe8 (Δe8) transcripts. Lower panels: RT-PCR results by the indicated primers of the cortex and MEF samples from WT (+/+), heterozygote (+/Δe8) and homozygote (Δe8/Δe8) mice. (**C**) cDNA sequencing to validate exon 8 deletion in *Mcph1*-Δe8 mice. PCRs using primers located on *Mcph1* exon 6 and exon 11 produced a 1547-base-pair fragment in WT cDNA and a 419-base-pair fragment in Δe8 cDNA. Right lower panel shows a loss of exon 8 and the only junction between exon 7 and exon 9 of *Mcph1*-Δe8 transcripts. (**D**) Western blot analysis of the expression of MCPH1 protein in neurospheres derived from control (Con, wild-type) and *Mcph1*-Δe8 homozygous mice (Δe8/Δe8) using an anti-MCPH1 antibody that recognizes residues in the central domain of MCPH1. β-actin was used as a loading control.

**Figure 2 cells-11-02715-f002:**
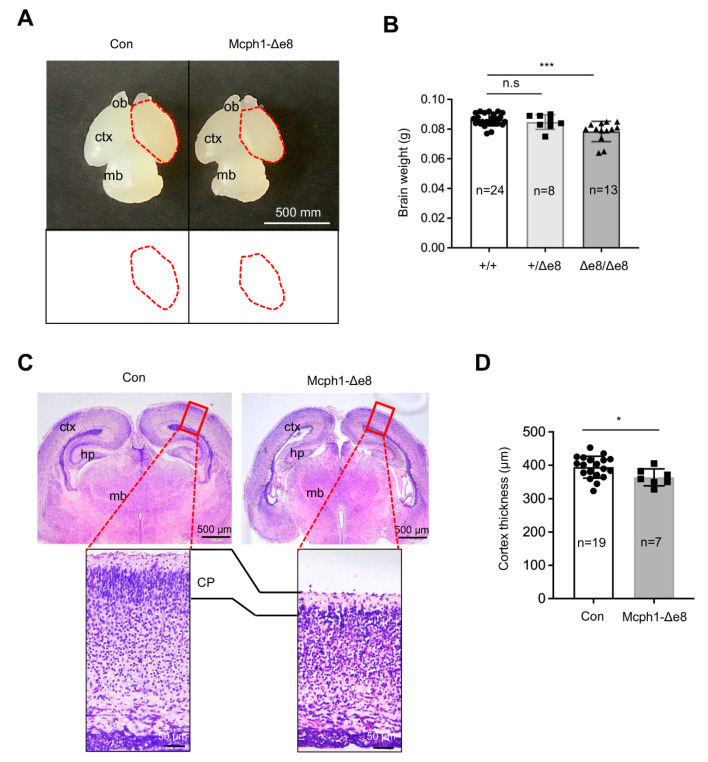
*Mcph1*-Δe8 mice develop primary microcephaly. (**A**) Macroscopic dorsal view of the mouse brain of wild-type control (Con) and homozygous *Mcph1*-Δe8 (Δe8/Δe8) mice at birth (P0). Schematic views of the cortical areas from control and *Mcph1*-Δe8 brains are shown in the lower panel. ob: olfactory bulb; ctx: cerebral cortex; mb: midbrain. (**B**) Quantification of the brain weights of the indicated genotypes of mice at P0. (**C**) H & E staining of coronal sections of control and *Mcph1*-Δe8 brains at P0. Overview of coronal brain sections and enlarged view of the forebrain cortex from the rectangular areas of the upper panel are shown. ctx: cerebral cortex; hp: hippocampus; mb: midbrain. (**D**) Quantification of the thickness of the cortex of control and homozygous Δe8/Δe8 mice. The cortex thickness of control: 394.22 μm and *Mcph1*-Δe8: 363.75 μm. *n*: the number of mice analyzed. Unpaired Student’s *t*-test was used for statistical analysis. *, *p* < 0.05; ***, *p* < 0.001; n.s, not significant.

**Figure 3 cells-11-02715-f003:**
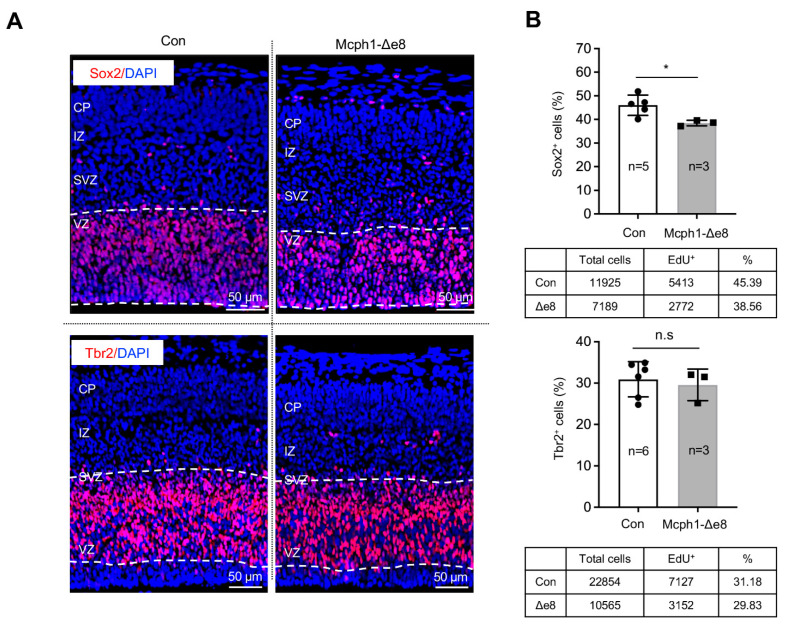
Reduction of neuroprogenitors in *Mcph1*-Δe8 embryonic cortex. (**A**) Immunofluorescence staining of sagittal sections of the E15.5 brain with Sox2 and Tbr2 antibodies. (**B**) Quantification of Sox2^+^ and Tbr2^+^ cells. *N*: the number of mice analyzed. The total numbers of cells of control (Con) and mutant (*Mcph1*-Δe8) scored are summarized under the respective graphs. Unpaired Student’s *t*-test was used for statistical analysis. *, *p* < 0.05; n.s, not significant.

**Figure 4 cells-11-02715-f004:**
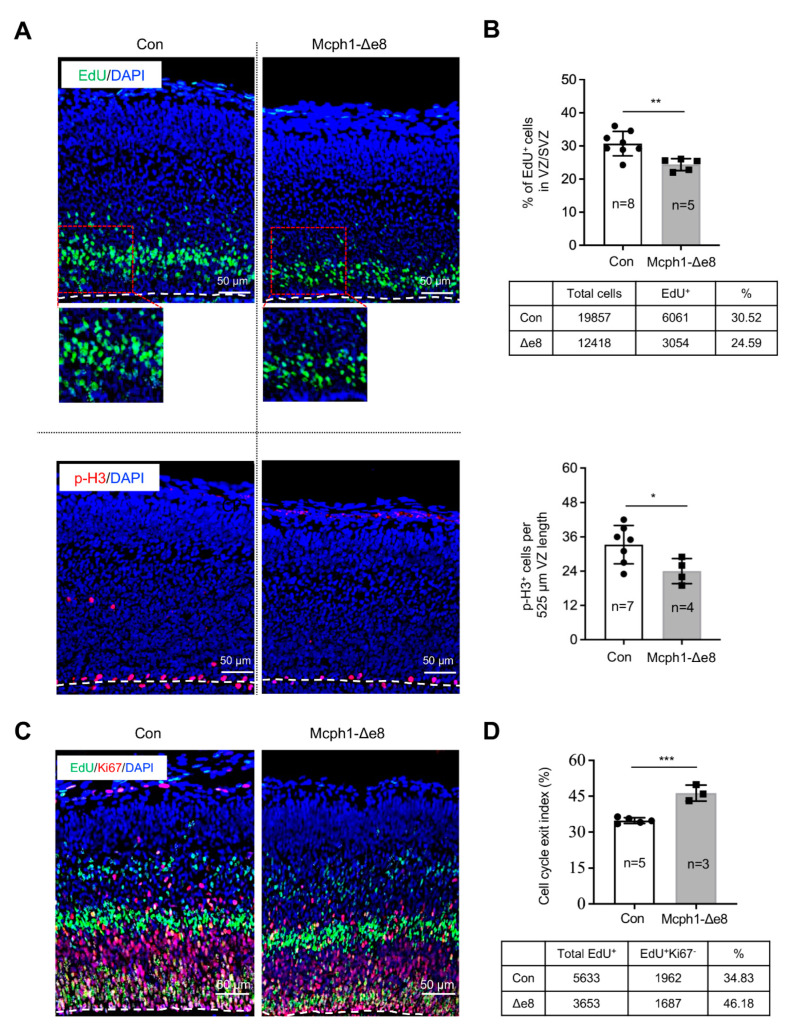
Proliferation defects of neuroprogenitors in the *Mcph1*-Δe8 embryonic cortex. (**A**) Immunostaining of the E15.5 embryonic cortex after EdU pulse labeling for 1 hr using antibodies against EdU (green) and pS28-H3 (a mitotic marker, red). The nucleus is counterstained by DAPI (blue). (**B**) Quantification of the percentages of EdU^+^ and pS28-H3^+^ cells among the total DAPI^+^ cells. (**C**) Double staining of the E15.5 brain cortex using antibodies against EdU (green) and Ki67 (red). EdU was injected on E14.5 of embryonic development and embryos were harvested at 24 hrs later (E15.5). (**D**) Cell cycle exit index was calculated by the ratio of EdU^+^Ki67^−^ vs. total EdU^+^ cells. The numbers of control (Con) and mutant (*Mcph1*-Δe8) cells scored are indicated in the tables below their respective graphs. *n*: the number of mice analyzed. Unpaired Student’s *t*-test was used for statistical analysis. *, *p* < 0.05; **, *p* < 0.01; ***, *p* < 0.001.

**Figure 5 cells-11-02715-f005:**
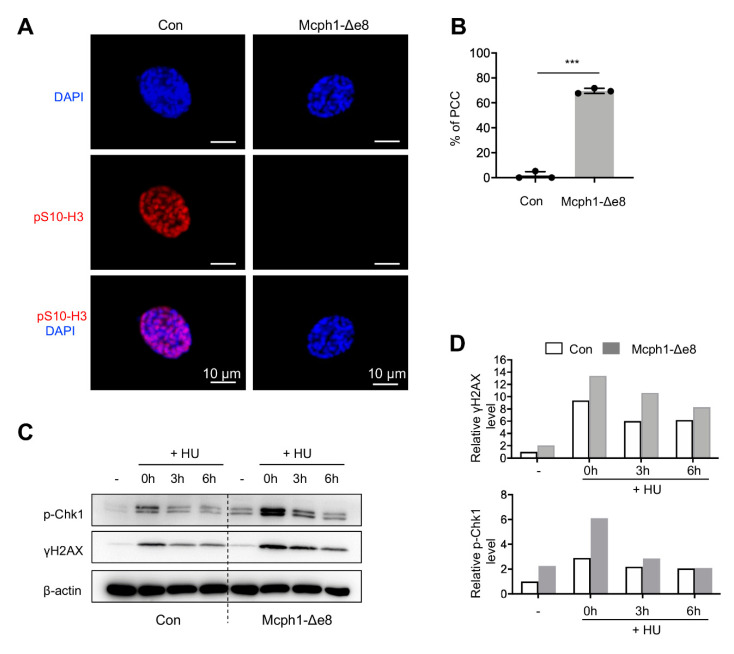
*Mcph1*-Δe8 cells exhibit PCC and defective DDR. (**A**) Representative images of PCC in *Mcph1*-Δe8 MEFs. Primary MEF cells were stained with a pS10-H3 antibody (red) and counterstained with DAPI (blue). (**B**) Quantification of the percentages of PCC cells (prophase cells lacking the pS10-H3 signal) in control and *Mcph1*-Δe8 primary MEF cells. More than 250 prophase cells were scored in each group of the indicated genotype. (**C**) Western blot analysis of p-Chk1 and γH2AX in control and *Mcph1*-Δe8 MEFs with or without HU treatment. β-actin was used as a loading control. (**D**) The quantification of the indicated protein intensities from panel (**C**). Unpaired Student’s *t*-test was used for statistical analysis. ***, *p* < 0.001.

**Figure 6 cells-11-02715-f006:**
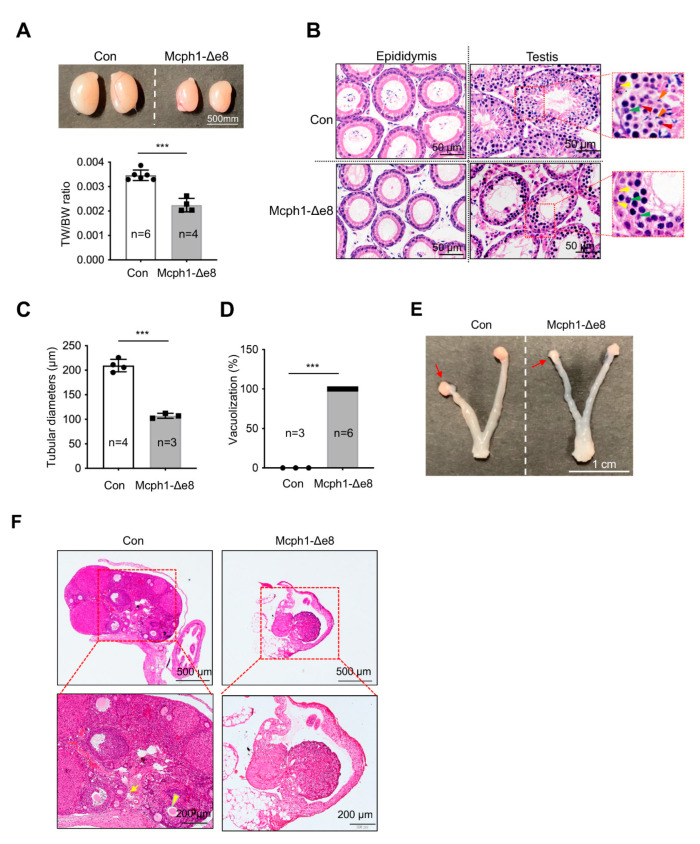
Defective gonad development in *Mcph1*-Δe8 mice. (**A**) Macroscopic view of testes from a control and a *Mcph1*-Δe8 mouse (6 weeks of age). The ratios of the testis weight (TW) to body weight (BW) from 6-week-old control and *Mcph1*-Δe8 mice are shown in the lower panel. (**B**) H&E staining of testis sections from 6-week-old control and *Mcph1*-Δe8 male mice. The epididymis (left panel) and seminiferous tubules (right panel) are shown. Yellow, green, red and blue arrowheads mark spermatogonia, pachytene spermatocytes, round spermatids and elongated spermatids, respectively. (**C**) Quantification of the diameters of seminiferous tubules of *Mcph1*-Δe8 male mice at P100. (**D**) Quantification of tubules lacking spermatocytes (vacuolization) as a percentage of the total tubules of 6-week-old *Mcph1*-Δe8 male mice. (**E**) Representative image of ovaries from a 6-week-old control and a *Mcph1*-Δe8 mouse. Arrows mark ovaries. (**F**) H&E staining of ovary sections of 6-week-old control and *Mcph1*-Δe8 mice. The yellow arrow and arrowhead point to a primary and secondary follicle, respectively. *n*: the number of mice analyzed. Unpaired Student’s *t*-test was used for statistical analysis. ***, *p* < 0.001.

## Data Availability

All datasets presented can be requested from the corresponding authors.

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
