# Peer review of "The Central Domain of MCPH1 Controls Development of the Cerebral Cortex and Gonads in Mice"

_cells, 2022, doi:10.3390/cells11172715_

Round 1
Reviewer 1 Report
Genetic studies have shown that mutations in the N-terminal and central domains of MCPH1 are strongly associated with microcephaly, indicating that these regions are important for brain development. The central domain of MCPH1 which is mainly encoded by exon 8, interacts with E3 ligase βTrCP2 and regulates G2-M transition of the cell cycle. However, the physiological function of this domain is not clear.
In this study, Wang et al. construct a mouse model with the deletion of the central domain of Mcph1 by deleting its exon 8 (Mcph1-Δe8). Mcph1-Δe8 mice exhibit reduced brain size and thinner cortex. Although the study is short of more detailed molecular mechanism, the authors have provided solid evidence that the central domain of Mcph1 is essential for its function during brain, ovary and gonad development in mammals which are caused by defect in self-renewal and premature differentiation of neuro progenitor cells during corticogenesis. In addition, Mcph1-Δe8 mice are sterile due to loss of germ cells in testis and ovary. All these findings indicate that Mcph1-Δe8 mice are similar to Mcph1 complete knockout and Mcph1-ΔBR1 mice.
Major point
Figure 5 which would show that Mcph1-Δe8 mouse embryonic fibroblasts exhibit the premature chromosome condensation is missing in the ms.
Minor points
1. Control in the figures should be labelled as either “control or Con”, not “Co”.
2. Significantly increased cell cycle exit index in Mcph1-Δe8 brain was shown in Figure 4. Could you explain why there was no increase of the Tbr2 positive cells?
Reviewer 2 Report
In this study, the authors found that the central domain of Mcph1 controls the development of the cerebral cortex and gonads in a mouse model, linking this domain with maintaining neuroprogenitor cells. This is a novel discovery that highly contributes to elucidating the basis of primary microcephaly, but the molecular mechanism underlying the functions of the central domain of Mcph1 remains unclear.
In summary, the authors concluded that the Mchp1 central domain is necessary to avoid the primary microcephaly phenotype. How do they reach such a conclusion? This study, together with the previous studies they quote, drives the reader to think that full protein is necessary to avoid it. Although the central domain is playing an important role, is not the only factor regulating primary microcephaly phenotype, as they imply.
Besides, the article proves some physiological phenotypes in a descriptive manner, it has no biochemical mechanisms attached to them.
Some comments are below:
Which proteins are the counter partners of Mcph1 regulating this process? Does Mcph1 lacking the central domain still interact with E3 ligase βTrCP2? Is Mcph1 being ubiquitinated by E3 ligase βTrCP2? Co-immunoprecipitation experiments are highly recommended.
A more detailed discussion about the role of the different Mcph1 domains in primary microcephaly and their functions is needed. Comparing these findings with the different human mutations reported to date is highly desirable and would make the discussion more solid.
I find the supplementary information provided little: a table with information about antibodies used, and a table of statistics (with all N numbers, cells analyzed, replicates, mean and SD, p-values, etc. for each figure) are missing.
Also, the statistical analysis is not well described (blinding, Size of sample, reproducibility).
The abstract is a bit too long and overlaps with the introduction.
In material and methods, image processing and image analysis processing is not described. How the image analysis was performed?
Excess of use of abbreviations without previous clarification (for example line 315 HU).
Specific comments:
Line 22. The mentioned central domain of Mcph1 interacts with E3 ligase βTrCP2 regulating G2-M transition. In which context this interaction takes place? which model/cell lines?
Line 53. How is related to cell cycle checkpoints? Experiments testing the proficiency of G2/M cell cycle checkpoints are highly recommended in their model using MEF cell lines.
Lines 86-87. Overlapping between abstract and introduction.
From line 195 – Generation of the mice: I find this part too methodological, lacking significance as a part of the results.
Figure 2A. Measurements of the area of the ROI (region of interest) in the cortex can be performed and plotted. Measurement in other areas could be included. Interestingly, the size of the mind-brain (mb) seems to be affected too.
Figure 2C. How the quantification of the cortex thickness is performed? which criteria are used in order to select the area that is going to be measured?
Figure 3A. Again, measurements of the area in this section can be measured.
There is enough space in the Figure to make a zoom-in into the areas of interest and show a magnification of them. The barplots show the “n” number that corresponds with the mice that were analyzed. However, raw data showing the total number of cells that were analyzed in each mouse, or how many sections in each mouse, are also highly desirable.
Line 270. Missing references or explanation (cleaved-Caspase3).
Line 274. How to be sure that is not the speed of replication that is affected? A reduction in the ratio of EdU-positive cells could be related to an increase in the speed of replication. In order to reach their conclusion, it is important to show that the replication speed is not affected. For example, performing live-cell experiments monitoring the S phase in their model using MEF.
Line 279. Missing references to other studies for this part.
Figure 4B. The number of analyzed mice for mutants (Mcph1 central domain deletion) is considerably reduced compared with control. I recommend increasing the number of analyzed mice to reduce effect size artefacts in the measurements. The representative images selected do not show a big difference, and the bottom part of the image looks cropped. Zoom-in and magnification of the area of interest can be included in order to illustrate differences better.
Line 299. Lacking background and references.
Line 311. Comparison of % of PCC between different models?
Line 315. HU (Hydroxyurea treatment). Lack of reasoning and justification based on background. A brief introductory sentence explaining why this experiment was performed would improve the manuscript.
Line 318. DDR malfunction in which sense? Does also affect the progenitor cells in the gonads?
Figure 5 is missing.
Figure 6F. what about the effect on female gonads? Any kind of quantification is missing.
Line 358. Overlapping with introduction
About the discussion:
Are both Mcph1 domains (N-terminal and central) necessary to avoid microcephaly phenotype?
Which domain plays a crucial role in this disease? What happens in the model lacking the N-terminal domain? how do they conclude that the reported effects are just due to the central domain? It is important to clarify this in the discussion.
The role of MCPH1 in cell cycle checkpoints (specifically G2/M transition) by interacting with other proteins like Chk1 and Plk1 has not been spotted at all. A further discussion about the misregulation of cell cycle checkpoints when Mcph1 lacks its central domain is necessary. To test the proficiency of some cell cycle checkpoints using the mentioned MEF cell lines could improve the knowledge about the role of this central domain and increase the quality of the study by adding some molecular mechanisms to it.
Round 2
Reviewer 2 Report
The authors have addressed many of the concerns raised and increased the credibility of the presented findings. I do not have any additional concerns.